# Effects of Maternal Environment on Seed Germination and Seedling Vigor of *Petunia* × *hybrida* under Different Abiotic Stresses

**DOI:** 10.3390/plants10030581

**Published:** 2021-03-19

**Authors:** Chi D. Nguyen, Jianjun Chen, David Clark, Hector Perez, Heqiang (Alfred) Huo

**Affiliations:** 1Mid-Florida Research Center, IFAS-University of Florida, Apopka, FL 32703, USA; chi285@ufl.edu (C.D.N.); jjchen@ufl.edu (J.C.); 2Department of Environmental Horticulture, IFAS-University of Florida, Gainesville, FL 32603, USA; geranium@ufl.edu (D.C.); heperez@ufl.edu (H.P.)

**Keywords:** petunia, maternal effect, heat, salinity, drought, ABA

## Abstract

Seed germination and seedling vigor can be affected by environmental cues experienced by the mother plant. However, information about how the maternal environment affects seed quality is scarce in ornamental plants. This study aimed to investigate the effects of two different maternal environments on the seed germination and seedling vigor of *Petuni*a × *hybrida* under a variety of abiotic stresses. Petunia mother plants were grown in either a greenhouse during the summer months or an indoor controlled-temperature-and-light environment. Collected seeds were subjected to external stressors, including polyethylene glycol (PEG), sodium chloride (NaCl), high temperature, and abscisic acid (ABA), to determine seed germination percentage and seedling vigor. Results indicated that seeds harvested from the mother plants grown in a controlled environment germinated better than seeds harvested from the mother plants grown in the greenhouse when suboptimal germination conditions were applied. Additionally, the seedlings from the controlled maternal environment performed better in both ABA and salinity stress tests than the greenhouse seedlings. Interestingly, the greenhouse seedlings displayed less reactive oxygen species (ROS) damage and lower electrolyte leakage than the controlled environment seedlings under dehydration stress. The difference in germination and seedling vigor of seeds from the two different maternal environments might be due to the epigenetic memory inherited from the mother plants. This study highlighted the strong impact of the maternal environment on seed germination and seedling vigor in Petunia and may assist in high-quality seed production in ornamental plants.

## 1. Introduction

Many ornamental and landscaping plants start their life as seeds, and high-quality seeds are essential for successful plant establishment. It is well known that the mother plant greatly influences seed traits, such as seed size, germination rate, and viability [1,2,3]. Specifically, the environmental cues, including light, temperature, soil moisture, and nutrients, the mother plant encounters can cause variations in seed quality even within the same genotype [4,5,6,7,8,9]. The resulting seed quality variation derived from the maternal effect can also shape the phenotypic plasticity of the progeny [1,10]. The benefits of growing the mother plant in favorable environments to obtain higher-quality seeds are well documented and practiced in many crop species, including tomatoes and legumes [2,11,12]. However, information about the effects of the maternal environment on the seed and seedling vigor of ornamental plants under abiotic stresses remains scarce.

The wholesale value of bedding and garden plants totaled $3.62 billion in the United States for 2018 and that of potted flowering plants for indoor or patio uses $877 million [13]. Petunia is among the top five most popular bedding and potted flowering plants due to its exquisite colors and the plethora of flowers. Additionally, petunia hybrid W115, also known as Mitchell, is commonly used as a model plant for analysis of molecular mechanisms underlying plant development. The importance of petunia in the floriculture industry and as a model plant species makes petunia an ideal choice for testing the maternal environment effects on seed quality. To ensure high seed quality, commercial seed companies generally produce seeds in regions with favorable light and temperature for the parental plants [11]. This practice restricts the time and location for seed production, which may hinder the production output of many plant species. For example, in tropical climate regions, where temperature and humidity are relatively high, flowering plants are more susceptible to diseases that negatively affect their seed quality. Alternatively, mother plants can be grown in a controllable environment to optimize temperature and light conditions for seed production. This method could be feasible for many compact plant species that produce seeds without requiring a large space, such as petunia and many other bedding and potted flowering plants.

All plant species require an optimum temperature range for their growth, and temperatures above or below the range may negatively impact plant physiological functions, leading to poor performance and productivity. Therefore, the increasing temperature due to global warming is a universal concern for the seed production industry [12,14,15]. In some cases, high-temperature stress during seed maturation can reduce germination due to the alteration of seed storage compounds [5]. Several reports have demonstrated that the accumulation and distribution of carbohydrates, oil, starch, and protein can be altered by high temperatures, resulting in great variation in seed quality and progeny development [14,15]. Light is another critical environmental factor, especially the photoperiod that can affect flowering time and seed production [16,17]. In addition to the quantity of light, the quality of light can regulate numerous physiological processes throughout the life cycle of plants. For instance, red light promotes seed germination, while far-red light delays germination by regulating abscisic acid (ABA) biosynthesis genes in barley [18]. Recent evidence shows that shading during seed development causes alterations in the biosynthesis of pro-anthocyanidins, fatty acids, and phytohormones and promotes subsequent germination in soybean [19]. Since both light and temperature are essential environmental cues, the present study aimed to understand the effects of two different maternal environments on petunia seed germination and seedling vigor under abiotic stress. The results could shed light on the influence of the maternal environment on seed production for the agricultural industry and will provide evidence in favor of using a controlled environment to produce high-quality petunia seeds.

## 2. Results

### 2.1. Petunia Seed Size and Seed Germination under Different Types of Abiotic Stress

Petunia seeds harvested from plants grown in the tent were significantly larger (~15% on average) than seeds harvested from the greenhouse plants (Figure 1). Seeds from plants grown in both maternal environments successfully germinated within three to four days on 1% agar control plates and reached their maximum germination of ~100% on the seventh day. However, seeds from the greenhouse plants exhibited more sensitivity to ABA treatment than seeds from the tent plants. The germinability of seeds from the greenhouse plants dropped to 61%, 17%, and 11% on 1, 1.5, and 2 µM of ABA-infused plates, respectively (Figure 2A,C–E). In contrast, seeds from the tent plants could germinate significantly better under the same ABA treatments. For example, almost all seeds from the tent plants were able to germinate on 1 µM ABA plates (Figure 2A,C), and 70% and 49% of the seeds germinated on 1.5 and 2 µM ABA plates, respectively (Figure 2A,D,E).

For the dehydration stress test, the water potential of −0.7 MPa did not significantly inhibit the germination of both greenhouse and tent seeds. However, germination of greenhouse seeds drastically decreased to 55% at −1.2 MPa, while 98% of tent seeds were able to germinate at this lower water potential (Figure 3A–C). Similar results were observed for salinity stress treatments. Nearly 100% of the seeds from both maternal environments germinated on agar plates containing 200 mM NaCl (Figure 3A,D). In contrast, only 27% of greenhouse seeds germinated on plates with 250 mM NaCl, while 73% of tent seeds successfully germinated under the same treatment (Figure 3A,E). Interestingly, seeds from plants grown in the tent displayed a higher percentage of germination at 36 °C and 38 °C (Figure 3A,F,G). Seeds from tent plants exhibited an average of 96% germination, compared to 88% for seeds from greenhouse plants (Figure 3A,F) at 36 °C. However, a significant difference in germination between the seeds from the two maternal environments was observed at 38 °C, where greenhouse seeds were not able to germinate at all, while 31% of the seeds harvested from tent plants were able to germinate at the same temperature (Figure 3A,G).

### 2.2. Post-Germination Growth under ABA or NaCl Stress

Seedlings grown on Murashige and Skoog (MS) medium with no stress treatment for 14 days displayed similar root (~70 mm) and shoot (~21 mm) lengths and leaf numbers (~6) regardless of the maternal environment the seeds originated from (Figure 4A–C). However, the root and shoot lengths of the greenhouse seedlings grown on a medium with 100 mM of NaCl decreased to 33 mm (~46%) and 9 mm (~42%), respectively, when compared to the control MS medium treatment (Figure 4A,B). In contrast, the root and shoot lengths of the tent seedlings decreased to 48 mm (~32%) and 17 mm (~19%), respectively, in comparison to the control MS medium treatment (Figure 4A,B). Greenhouse seedlings under salinity stress also displayed one less leaf on average than the tent seedlings (Figure 4C). These results suggested that seedlings from seeds harvested from plants under different maternal environments significantly differ in their responses to salinity stress. Similar results were also observed for ABA treatment. Although seedlings from both maternal environments on MS medium supplemented with 1 µM ABA had the shortest root length when compared to the control MS medium and NaCl treatment, the greenhouse seedlings were more sensitive to ABA treatment than the tent seedlings. The average root length of the greenhouse seedlings was 34 mm, which was 6 mm shorter than that of the tent seedlings under ABA treatment (Figure 4A,B). In contrast, the shoot length difference between plants from the two maternal environments was only 2 mm, and the greenhouse seedlings also had one less leaf than the tent seedlings on average under ABA stress (Figure 4B,C). While all measured parameters were significantly different between the two maternal environments under both ABA and NaCl treatments, stronger responses to salinity stress were found in comparison to ABA stress (Figure 4B,C).

### 2.3. Alterations in the Reactive Oxygen Species (ROS) and Electrolyte Leakage of Petunia Seedlings under Different Types of Abiotic Stress

Staining of leaves with 3,3′-diaminobenzidine (DAB) and nitrotetrazolium blue chloride (NBT) for detecting hydrogen peroxide (H_2_O_2_) and superoxide anion (O_2_ˉ) was carried out for each treatment to understand how abiotic stresses affect ROS homeostasis in petunia seedlings derived from two different maternal environments. A slight increase in H_2_O_2_ accumulation was observed when the seedlings were subjected to a short period (4 h) of heat stress (Figure 5B). A noticeable difference in H_2_O_2_ accumulation was observed in seedlings from the two maternal environments under dehydration treatment with PEG8000, where the staining for H_2_O_2_ in the greenhouse seedlings was visually lighter than that in the tent seedlings (Figure 5C). Similarly, O_2_ˉ staining was also slightly less in the greenhouse seedlings under dehydration stress (Figure 5H). The staining of ROS was significantly darker in seedlings from both environments under salinity stress when compared to the control MS medium, but no significant difference was observed between the seedlings derived from the two maternal environments (Figure 5A,D,I). Although there was a distinct difference in the petunia seedlings’ growth under 1 µM ABA stress between the two maternal environments (Figure 4A–C), no significant visual difference in H_2_O_2_ and O_2_ˉ staining was found in seedlings from plants grown in the two maternal environments under ABA treatment (Figure 5E,J). In addition, H_2_O_2_ and O_2_ˉ accumulation remained relatively the same under ABA treatment when compared to the control MS medium treatment (Figure 5A,E,F,J).

Similar to ROS-staining results, there was a significant increase in electrolyte leakage when the petunia seedlings from both maternal environments were subjected to treatments of dehydration, salinity, and ABA in comparison to the control condition (MS medium only). However, there was no significant difference in the electrolyte leakage between seedlings under heat stress and the control MS medium. A significant difference was only observed between seedlings from the two maternal environments under dehydration stress. Where the tent seedlings on average had 7% more electrolyte leakage than the greenhouse seedlings (Figure 6).

## 3. Discussion

### 3.1. Petunia Seed Germination and Seedling Vigor under Different Types of Abiotic Stress

Seed germination is a critical phase in the plant life cycle and has biological, economic, and ecologic importance. The process of seed germination is regulated by endogenous hormonal cues and environmental signals that determine whether an imbibed seed remains dormant or completes germination [20,21]. When seeds are germinated under optimal conditions, uniform and high levels of germination are usually obtained, even when the seeds are from different sources. However, seeds harvested from plants grown in different maternal environments may have contrasting capabilities to germinate and establish plants under suboptimal germination conditions. As shown in Figure 2A and Figure 3A, seeds from both maternal environments have high germinability with no stress treatment but exhibit different responses under stressful conditions.

The phytohormone abscisic acid (ABA) plays a critical role in numerous biological processes, including key events during seed development, seed dormancy, and seedling growth [22]. It has been shown that dormant seeds continue to synthesize ABA during imbibition, while ABA catabolism is favored in seeds subjected to effective dormancy-breaking treatment [23]. In addition to seed dormancy, ABA serves as a signal in the response to environmental stresses, both biotic and abiotic, such as temperature, drought, and salinity [22]. For example, de novo ABA synthesis increases in seeds under unfavorable environmental conditions, resulting in secondary seed dormancy to prevent plant development in harsh conditions [24,25]. When petunia seeds were exposed to different stresses during imbibition, the germination percentage decreased accordingly (Figure 2 and Figure 3), probably due to the high level of de novo ABA production. However, seeds from mother plants grown in the tent environment exhibited less sensitivity to ABA, salt, dehydration, and high-temperature stresses, resulting in a higher percentage of germination compared to seeds from mother plants grown in a greenhouse environment (Figure 2 and Figure 3).

To determine whether seed vigor can affect seedling growth, germinated seeds were immediately subjected to ABA or NaCl treatment. Morphological characteristics, such as root length, shoot length, and the number of leaves, all indicated that the tent seedlings performed better than the greenhouse seedlings under both types of stress. Root and shoot lengths of the tent seedlings under salinity stress were exceedingly longer than those of the greenhouse seedlings (Figure 4A,B). The seedling response to salinity stress is most likely due to the inhibition of water uptake by NaCl, while ABA elicits multiple physiological responses other than water stress. In general, the structure of the root system will influence how the plant acquires water and nutrients, which, in turn, affects the shoot growth [26]. Thus, longer roots indicate the plant’s ability to seek water deep in the soil, particularly under stressful conditions. The ability to overcome suboptimal germination conditions as well as have strong seedling growth may be due to higher accumulation of reservoirs, such as sugars, organic acids, and amino acids, during seed maturation in the petunia seeds from the favorable tent environment (Figure 1). For example, there was a significant difference in the accumulation of γ-aminobutyric acid (GABA) in tomato seeds grown under different concentrations of nitrate and phosphate, which, in turn, affected the performance of seeds and seedlings (the interaction between the genotype and maternal nutritional environments affects tomato seed and seedling quality). Similarly, when tomato introgression lines were grown in the field with or without saline stress, maternal salinity stress imposed a transgenerational effect on the progeny seeds, especially in the accumulation of amino acids and sugars, which exhibited a negative effect on seed germination [27]. A great reservoir can provide adequate nutrients, especially in the early developmental stage without any photosynthesis, as observed in some vegetable species subjected to seed-priming treatments [28]. Although supplemental nutrients were provided in the MS medium, the difference in the original reservoir between the greenhouse seeds and the tent seeds may have caused different seedling growth rates under stressful conditions. As observed, the seeds from the tent environment had larger seed sizes and stronger vigor, which may have contributed to better germination and post-germination growth under a variety of abiotic stresses. Indeed, previous studies have demonstrated a close relationship between seed size and seed vigor in rice and barley [29,30,31]. The results in this study suggested that the maternal environment strongly affects petunia seed germination and subsequent seedling vigor and that higher-quality seeds can be produced by growing petunia mother plants in a controlled indoor environment at a constant 23 °C and under light-emitting diode (LED) light.

### 3.2. Petunia Seedling Reactive Oxygen Species (ROS) Accumulation and Electrolyte Leakage in Response to Different Types of Abiotic Stress

In plants, ROS are continuously produced as by-products of aerobic metabolism, predominantly in chloroplasts, mitochondria, and peroxisomes [32]. To prevent potential toxic effects or oxidative stress from over-accumulation of ROS, plants are generally able to remove or detoxify them by cellular anti-oxidative mechanisms [33,34]. Thus, ROS are dynamically generated or eliminated to maintain homeostasis for normal functions of plant cells, leading to plant adaption to different environments [32,35,36]. Although ROS are critical players in the stress signaling responses, disruption of ROS homeostasis generally causes oxidative stress and destruction of the cell membrane. This is evident in the high percentage of electrolyte leakage in petunia seedlings under dehydration and salt stress, where staining of ROS staining in seedlings from both environments was significantly darker in comparison to that in the control MS medium (Figure 5C,D,H,I and Figure 6).

However, ROS are not simply a toxic by-product but are involved in many plant physiological processes as signal molecules to trigger the reaction of cells to a specific stimulus. Interestingly, H_2_O_2_ and O_2_ˉ staining remained relatively the same as that of the control when petunia seedlings were under ABA stress (Figure 5A,E). Overwhelming evidence has suggested that H_2_O_2_ is a key component in the signaling transduction process associated with stress tolerance and that H_2_O_2_ priming can modulate abiotic stress tolerance [37,38,39,40]. ABA can also be used as a priming agent to increase abiotic tolerance, and aerobic metabolism may have been reduced to direct the energy toward protective mechanisms [41]. Therefore, the level of H_2_O_2_ was temporarily lowered, but a higher accumulation of H_2_O_2_ would be expected after prolonged exposure to a higher concentration of ABA, as seen in other studies [42,43].

The seedling vigor test immediately post-germination after a NaCl or ABA stress test indicates that strong seed vigor can positively influence seedlings’ growth under abiotic stress, as observed when the tent seedlings outperformed the greenhouse seedlings in root and shoot growth (Figure 4). Intriguingly, when abiotic stresses were applied to seven-day-old seedlings, the ROS and electrolyte leakage results indicated that the greenhouse seedlings were able to tolerate the dehydration test more than the tent seedlings. While a higher reservoir can influence seed vigor, lower endogenous ABA during seed development also contributes to seed germination and early seedling growth [22]. Based on our results, the seeds from the tent plants were less sensitive to ABA and abiotic stresses than the greenhouse seeds, implying a lowered level of endogenous ABA in the tent seeds. As a stress hormone responding to environmental stresses to protect the plant from ROS damage, higher endogenous ABA in the greenhouse seedlings may have influenced the ROS network during stress treatment [37]. Therefore, ROS accumulation was reduced in the greenhouse seedlings when compared to the tent seedlings when stressful conditions were applied. Despite many studies showing a significant effect of maternal environments on the performance of plant progenies, the molecular mechanism underlying maternal regulation is not fully explored. Recently, an increasing number of reports have demonstrated that epigenetic memory could be inherited from mother plants that experienced environmental stress and that these memory markers may enable plant progenies to evoke stronger and more rapid responses to combat similar stresses [44,45,46]. Whether the maternal effect on petunia seedlings in this study is epigenetically regulated remains to be clarified in the future.

## 4. Materials and Methods

### 4.1. Plant Materials and Maternal Environments

Seeds of petunia hybrid W115 were collected from the greenhouse and stored at the University of Florida’s Mid-Florida Research and Education Center in a dark, cool room at a constant temperature of 10 °C with 30% relative humidity for three months. Seeds were germinated on filter paper moistened with water, and germinated seeds were grown in 5 cm pots under fluorescent white light in a 16 h/8 h light/dark photoperiod at 25 °C for two weeks. The seedlings were then transplanted into 3.78 L pots filled with garden soil and moved to either the greenhouse or an indoor tent (iPower GLTENTM1) and left there from the beginning of May to the end of July 2020. The steel greenhouse covered with polyethylene (PE) was located at 28.6° N, 81.5° W, with a temperature range from 22 °C to 31 °C and an average photoperiod of 14 h of daylight. The plants in the greenhouse were drip-irrigated twice a day for 15 min, and a similar amount of Osmocote plus 15-9-12 fertilizer was applied to all the plants. In contrast, the tent was placed in an air-conditioned room maintained at 23 °C with 60% relative humidity, and the plants were watered, as needed, to keep the soil moist. The same fertilizer was applied to the plants in the controlled environment as the greenhouse plants. The plants in the tent were under light-emitting diode (LED) light with a full spectrum from 380 to 735 nm provided by Heliospectra RX30 (Heliospectra, Chicago IL USA) for 2 h, interrupted by 2 h of only red (660 nm) and far-red (735 nm) light, and followed by 12 h of full-spectrum light, for a total of 16 h of light and 8 h of darkness. Five plants were randomly assigned per replication, and the experiment was repeated three times for each of the maternal environments. The two maternal environments henceforth are designated as greenhouse and tent environments to simplify the description.

### 4.2. Seed Size Measurement

Seeds collected from the plants grown in the two maternal environments were air-dried in a paper envelope at room temperature for three weeks before they were stored in a cool, dark room at 10 °C and 30% relative humidity. After one month, 25 seeds from each maternal environment were evenly placed on moistened filter paper and incubated overnight at room temperature (25 °C) (Figure 1). Seed images were captured using a camera with the same magnification, and ImageJ was used to determine the seed area [47]. The experiment was repeated six times for each maternal environment.

### 4.3. Seed Germination under Different Types of Abiotic Stress

Seeds were surface-sterilized in 90% alcohol for 1 min, then submerged in a 1.3% hypochlorite solution, and continuously shaken at 160 rpm for 10 min. Afterward, the seeds were washed with sterilized water seven times before being placed on sterilized filter paper to dry. Each seed was moved individually to different treatments and arranged as shown in Figure 1. Seeds from both maternal environments were tested on a 1% agar plate as the control or on a 1% agar plate infused with stressors to mimic different abiotic stresses. A germination test was carried out under a 16 h/8 h fluorescent-white-light/dark photoperiod at room temperature (25 °C), except for the high-temperature treatment. Seeds with radicle protrusions of at least 1 mm were considered germinated.

Four types of abiotic stress were tested in this study. For the abscisic acid (ABA) stress test, a stock solution of 100 mM was used to make dilutions of 1, 1.5, and 2 µM ABA after the temperature of autoclaved media was reduced to 50 °C. To mimic dehydration stress, PEG8000 (Fisher Scientific, Waltham MA USA)-infused plates were prepared following the protocol described by van der Weele et al. [9]. Briefly, 400 and 550 g/L of PEG8000 were dissolved in water before overlaying the PEG solution onto agar plates. The agar-to-PEG overlay solution volume ratio was kept at 2:3 to create a water potential of −0.7 and −1.2 MPa, respectively. The PEG8000 solution was left on the agar plate overnight to achieve equilibrium and discarded before placing the seeds onto the agar. For the salinity stress test, seeds were germinated on agar plates containing either 200 or 250 mM sodium chloride (NaCl). For high-temperature stress, seeds were tested in a germination chamber at a constant 36 °C or 38 °C (Fisher Scientific Waltham MA, USA). The percentage of germination was calculated by dividing the number of germinated seeds by the total number of seeds per replication, and pictures were taken after seven days of imbibition. Each treatment plate contained 25 seeds from tent plants and 25 seeds from greenhouse plants; six replications were applied to each treatment.

### 4.4. Seedling Vigor Assay

Sterilized seeds were first germinated on a 1% agar plate, and seeds with radicles of at least 1 mm were moved onto growth plates containing NaCl or ABA. Briefly, six germinated seeds from each maternal environment (greenhouse or tent) were placed on a growth medium (Murashige and Skoog (MS) + 3% sucrose + 1% agar; pH 5.8), with the addition of 100 mM NaCl or 1 µM ABA as stress treatment, while a growth medium without ABA or NaCl was used as the control. These growth media plates were placed vertically, and the seedlings were grown under the same fluorescent white light and photoperiod as previously mentioned. Six replications were applied to each treatment, with a total of 36 seedlings per maternal environment. Root length was measured from the base of the shoot to the longest part of the root, and shoot length was measured from the base of the shoot to the top of the shoot with a standard ruler. The total number of leaves (including cotyledons) was counted. All measurements and pictures were taken after 14 days of treatment.

### 4.5. Seedling Stress Test: Electrolyte Leakage Measurement and Reactive Oxygen Species (ROS) Staining

Germinated seeds on 1% agar were first grown on a growth medium for seven days. Afterward, 4 seedlings from each maternal environment (greenhouse or tent) were moved onto a growth medium containing PEG8000 (−0.7 MPa), 100 mM NaCl, and 1 µM ABA for 10 days before measuring ROS and electrolyte leakage. Only for high-temperature stress, 17-day-old seedlings on a growth medium were moved to a growth chamber at a constant temperature of 40 °C for 4 h before measuring ROS and electrolyte leakage. All media preparations were the same as the ones for the seed germination test, except that 40 mL of each medium was poured into 473 mL polypropylene deli containers (Fabri-Kal, Kalamazoo MI, USA) to give space for seedling growth. Five replications were applied to each treatment, for a total of 20 seedlings per maternal environment for each treatment.

For electrolyte leakage determination, seedling shoots from each treatment were incubated in 25 mL of deionized water while shaking overnight at 70 rpm on a platform shaker (Innova 2100, New Brunswick Scientific, Enfield CT USA) at room temperature (25 °C). Relative electrolyte leakage was measured according to the method of Cao et al., with slight modifications based on the description by Ilik et al. [48,49]. The initial electrical conductivity (EC_i_) was measured with a conductivity meter (Model A112, Thermo Scientific, Waltham MA USA), followed by autoclaving the samples for 10 min and cooling to room temperature to measure the total electrical conductivity (EC_t_). The percentage of electrolyte leakage was expressed as (EC_i_/EC_t_) × 100.

Histochemical detection of the reactive oxygen species (ROS) hydrogen peroxide (H_2_O_2_) was performed using the methods described by Daudi et al., and the method described by Kumar et al. was adopted for measuring superoxide (O_2_ˉ) anion accumulation in petunia seedlings. Briefly, a solution containing 1 mg/mL of 3,3′-diaminobenzidine (DAB) (Sigma Aldrich, Saint Louis MO USA), 0.05% *v*/*v* of Tween 20, and 10 mM of Na_2_HPO_4_ was used to detect hydrogen peroxide (H_2_O_2_), and 0.2% of nitrotetrazolium blue chloride (NBT) in 50 mM sodium phosphate buffer (Sigma Aldrich, Saint Louis, MO, USA) was used for superoxide (O_2_ˉ) anion staining. The seedlings were incubated in the staining solutions overnight at room temperature. Then the solutions were discarded and replaced with absolute ethanol and heated in a boiling water bath for 10 min to remove all chlorophyll content. The stained seedlings were preserved with 60% glycerol, and they were immediately spread out on paper for imaging with a camera. From each maternal environment and each treatment, 12 seedlings were used for detecting each type of ROS.

### 4.6. Statistical Analysis

The data were analyzed using Student’s *t*-test. All statistical analyses and graphs were produced using R Studio [50].

## Figures and Tables

**Figure 1 plants-10-00581-f001:**
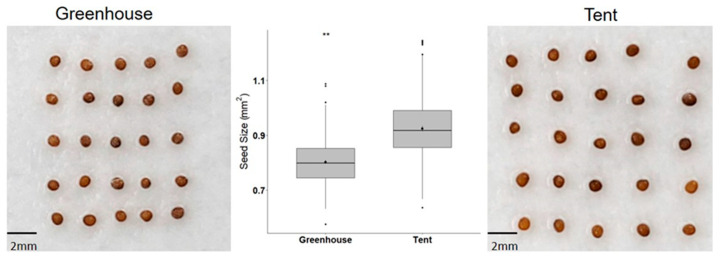
Size of petunia seeds from plants grown in the tent and the greenhouse. Pictures were taken at the same magnification, and ImageJ was used to determine the seed size. The box plots represent the mean ± standard deviation. ** Significant difference between seeds from tent and greenhouse plants at *p* < 0.01 (*t*-test). *n* = 150 seeds per maternal environment.

**Figure 2 plants-10-00581-f002:**
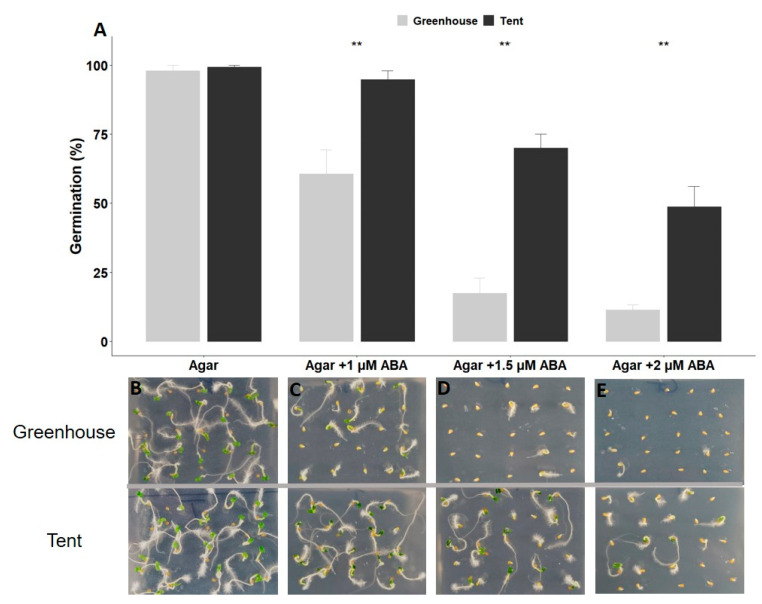
Germination response of petunia seeds from greenhouse or tent plants to abscisic acid (ABA) treatment. (**A**) Germination percentage of seeds from plants grown in two maternal environments at different concentrations of ABA. (**B**) Representative images of seed germination on a 1% agar plate without ABA or with different concentrations of ABA (**C**–**E**). The percentage of germination was recorded after a 7-day imbibition. Six replications, with 25 seeds per replication (*n* = 150 total), from greenhouse or tent plants were applied to each treatment. The vertical bars represent the mean ± standard error. ** Significant difference between seeds from greenhouse and tent plants under the same germination condition at *p* < 0.01 (*t*-test).

**Figure 3 plants-10-00581-f003:**
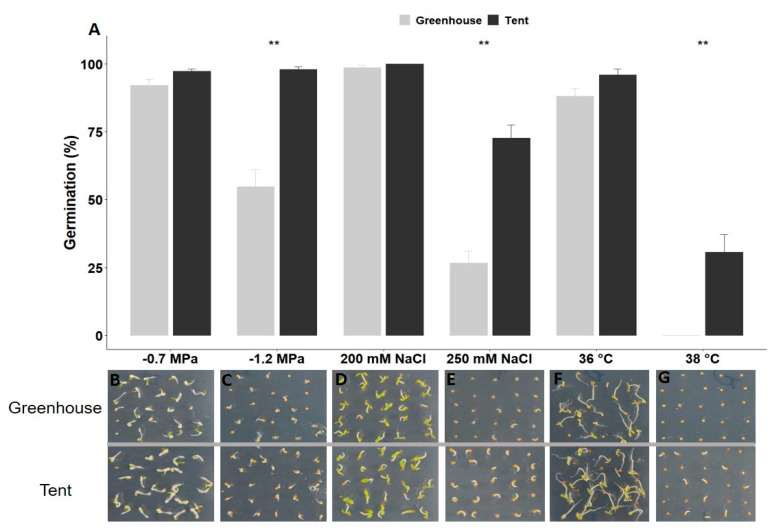
Germination response of petunia seeds from greenhouse or tent plants to dehydration, salinity, and high-temperature treatment. (**A**) Germination percentage of seeds from greenhouse or tent plants under polyethylene glycol (PEG), NaCl, and high-temperature treatment. (**B**–**G**) Representative images of seed germination on a 1% agar plate infused with PEG8000 (**B**,**C**), with NaCl (**D**,**E**), or at different high temperatures (**F**,**G**). The percentage of germination was recorded after a 7-day imbibition. Six replications, with 25 seeds for each replication (*n* = 150 total), from greenhouse or tent plants were applied to each treatment. The vertical bars represent the mean ± standard error. ** Significant difference between seeds from greenhouse and tent plants under the same germination condition at *p* < 0.01 (*t*-test).

**Figure 4 plants-10-00581-f004:**
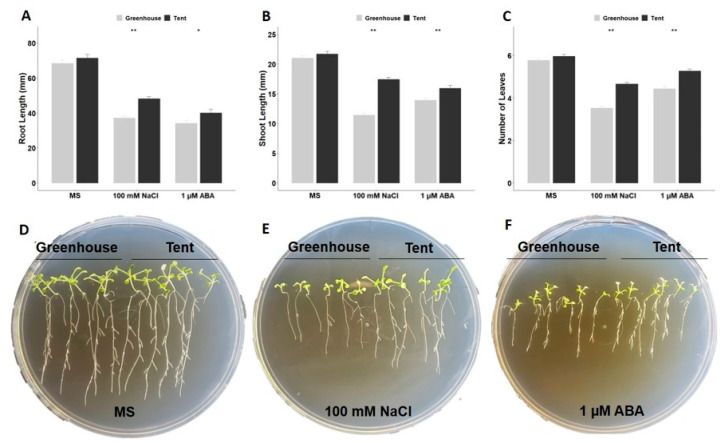
Effect of ABA and NaCl on the post-germination growth of petunia seeds harvested from greenhouse or tent plants. Uniformly germinated seeds were moved and placed on Murashige and Skoog (MS) medium, MS medium with 100 mM NaCl, or MS medium with 1 µM ABA. Root length (**A**), shoot length (**B**), and the number of leaves (**C**) for each treatment were measured after 14 days. (**D**–**F**) Representative image of seedlings’ growth after MS medium, 100 mM NaCl, and 1 µM ABA treatments, respectively. Six replications, with 6 seedlings for each replication (*n* = 36 total), from greenhouse or tent plants were applied to each treatment. The vertical bars represent the mean ± standard error. * and ** Significant difference between seedlings originating from plants grown in the greenhouse and those grown in the tent under the same treatment condition at *p* < 0.05 and *p* < 0.01 (*t*-test), respectively.

**Figure 5 plants-10-00581-f005:**
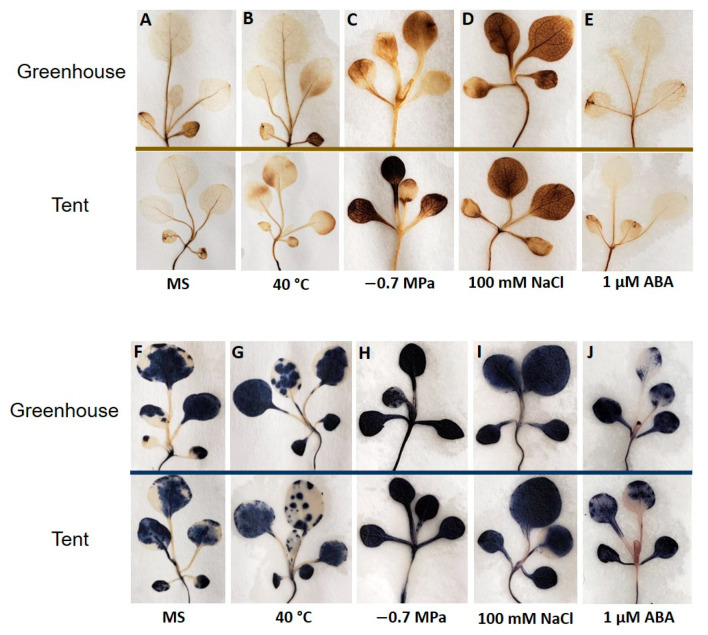
Alteration in reactive oxygen species (ROS) in petunia seedlings in response to different stress treatments. (**A**–**E**) Representative images of histochemical detection of hydrogen peroxide (H_2_O_2_) after treatment with high temperature, PEG8000, NaCl, and ABA. (**F**–**J**) Representative images of histochemical detection of superoxide anion (O_2_ˉ) after treatment with high temperature, PEG8000, NaCl, and ABA.

**Figure 6 plants-10-00581-f006:**
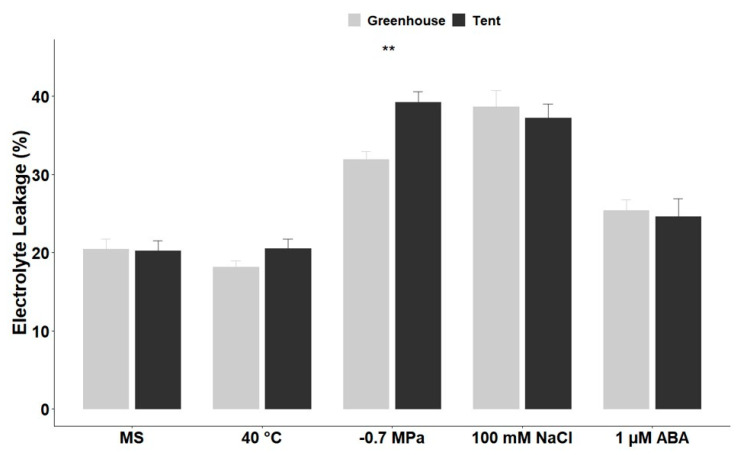
Electrolyte leakage percentage in petunia seedlings after treatment with high temperature, dehydration, NaCl, and ABA. Electrolyte leakage was measured after 10 days of treatment. Five replications with 4 seedlings for each replication (*n* = 20 total) from greenhouse or tent plants were applied to each treatment. The vertical bars represent the mean ± standard error. ** Significant difference between seedlings from seeds originating from plants grown in the greenhouse and the tent under the same treatment condition at *p* < 0.01 (*t*-test).

## Data Availability

All data generated or analyzed during this study are included in this published article.

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
