# Peer review of "Effects of Maternal Environment on Seed Germination and Seedling Vigor of Petunia × hybrida under Different Abiotic Stresses"

_plants, 2021, doi:10.3390/plants10030581_

Round 1

Reviewer 1 Report

This is a well-written paper describing the effects of the maternal environment in the initial stages from seed to seedling of Petunia × hybrida under different abiotic stresses.

1) This work is focused on an important topic in plant breeding: maternal effects in early plant stages between seed to seedling phase change. The physiological analysis is robust and has been adequately analyzed. One of the most interesting finds of this work is that seedlings from the controlled maternal environment performed well under ABA and salinity stress when comparing to the ones from the greenhouse seedlings. This result suggests that transcriptional regulation is important to control seed-to-seedling transition under these stresses. However, this work lacks functional results and further investigation is needed. Specifically, I suggest performing RT-qPCR of "key" genes of ABA biosynthesis, once authors discussed that de novo ABA synthesis has been shown to be increased in seeds under unfavorable environmental conditions, resulting in secondary seed dormancy to prevent plant development in harsh conditions. To this end, when authors discussed that the ability to overcome suboptimal germination conditions, as well as strong seedling’s growth, maybe due to higher accumulation of reservoirs, such as sugars, organic acids, and amino acids during seed maturation in the petunia seeds from the favorable tent environment. This is stating the obvious, there will be the initial activation of important metabolic processes during this phase. I would like to see more discussion around it or even an untargeted metabolite profile that would support the hypothesis. 

2) I would like to see a description of the point that the seed was considered germinated. Would it be at Radicle Protrusion? Reading through the manuscript I was confusing if it was considered germinated when you have cotyledons open. This would consider as the seedling establishment.  

3) The short discussion doesn't really set up the manuscript well. There should be some discussion about what happens in the seedling versus the seed, what would be seen with separating out the emergent root and shoot from the seed and contrasting the seed and emergent seedling. Also, some discussion about when the plants are producing carbon and acquiring nitrogen. A little more discussion around about the rapid decreases associated with the plant fully utilizing the seed stores and becoming self-sufficient under the stresses.

Author Response

Dear Reviewer 1,

Thank you for your insightful comments on the manuscript. Please find our responses to your specific concern below.  

1) This work is focused on an important topic in plant breeding: maternal effects in early plant stages between seed to seedling phase change. The physiological analysis is robust and has been adequately analyzed. One of the most interesting finds of this work is that seedlings from the controlled maternal environment performed well under ABA and salinity stress when comparing to the ones from the greenhouse seedlings. This result suggests that transcriptional regulation is important to control seed-to-seedling transition under these stresses. However, this work lacks functional results and further investigation is needed. Specifically, I suggest performing RT-qPCR of "key" genes of ABA biosynthesis, once authors discussed that de novo ABA synthesis has been shown to be increased in seeds under unfavorable environmental conditions, resulting in secondary seed dormancy to prevent plant development in harsh conditions. To this end, when authors discussed that the ability to overcome suboptimal germination conditions, as well as strong seedling’s growth, maybe due to higher accumulation of reservoirs, such as sugars, organic acids, and amino acids during seed maturation in the petunia seeds from the favorable tent environment. This is stating the obvious, there will be the initial activation of important metabolic processes during this phase. I would like to see more discussion around it or even an untargeted metabolite profile that would support the hypothesis. 

Response: We truly appreciate reviewer’s constructive and insightful suggestion. ABA is a well-known hormone that regulates seed germination and dormancy, the biosynthesis and signaling of ABA also respond to environmental cues to regulate seed germination plasticity. However, several other genetic factors such as Flowering Locus T and Delay of Germination 1 may play significant roles in environmentally mediated seed development (Penfiled and MaGregor, 2016). In this study, we tried to show how environmental changes may affect seed quality of petunia, which perfectly fall into the topic scope of this special issue “Seed Dormancy and Germination in Response to Climate Change”. The molecular mechanism of how plant hormones and genetic factors responds to environmental changes will be fully investigated in our separate project, as this reviewer recommend. Currently multiple genetic materials are under development, sophisticated experiments will be conducted to dissect the molecular mechanism underlying the environmental regulation of petunia seeds. Regarding of the discussion on the environmental effect on seed reservoirs, we have added a brief but insightful discussion in the discussion section as suggested by this reviewer (Line 245-253)

2) I would like to see a description of the point that the seed was considered germinated. Would it be at Radicle Protrusion? Reading through the manuscript I was confusing if it was considered germinated when you have cotyledons open. This would consider as the seedling establishment.  

Response: The description was added to the main manuscript (Line 350-351).

3) The short discussion doesn't really set up the manuscript well. There should be some discussion about what happens in the seedling versus the seed, what would be seen with separating out the emergent root and shoot from the seed and contrasting the seed and emergent seedling. Also, some discussion about when the plants are producing carbon and acquiring nitrogen. A little more discussion around about the rapid decreases associated with the plant fully utilizing the seed stores and becoming self-sufficient under the stresses.

Response: Given the limited length of this manuscript, we are not able to cover all discussion. The discussion of this manuscript is about 2 full pages long containing 49 references. Although this reviewer has provided insightful comments above, we do not agree to add such discussion as proposed in the Q3. This study focused on how maternal environment affects seed quality, in particular seed germination. The discussion on “when the plants are producing carbon and acquiring nitrogen” and on “rapid decreases associated with the plant fully utilizing the seed stores” is not related to our study, which may confuse readers of this paper. On the other hand, we accepted the suggestion on discussion on the how maternal environment affect seed reservoir and seed vigor (see our response to Q2 above) and provided a concise but informative discussion.

Reviewer 2 Report

The present paper is an original research paper and falls within the scope of the journal. Though numerous articles on seeds germination of ornamental plants have been already published, it presents new outcomes about the effects of two maternal environments on germination and vigor of Petunia seeds under different abiotic stresses. The manuscript is well written, with a good English grammar style.

The Abstract clearly summarizes the aim of the study and the main results obtained.

The Introduction clearly describes the state of art about the effects of maternal environments on growth of ornamental seedlings and the importance of growing site conditions (light and temperature). The aim of the study is clearly defined.

In my opinion, some sentence about the effect of abiotic stresses on seeed germination should be included in this section, before describing the scope of the experiment.

Results

This section clearly describes the main results obtained from the present experiment and regularly reported in the figures.

Discussion

In this section, the authors clearly try to understand the obtained results and to explain the effects of the different maternal enviroments on the studied parameters. The authors also try to justify the outcomes giving their personal hypothesis and comparing their results with those coming from previous studies.

Lines 244-245: modify as suggested “Great reservoir can provide adequate nutrients especially in the early developmental stage without any photosynthesis, as observed in some vegetable species subjected to seed priming treatments (31)”.

Please improve this section by adding other personal considerations on the effects of maternal enviroment on abiotic stress tolerance and other related references.

The Materials and methods clearly describe plant growth conditions and the analytical procedures followed for the determination of electrolyte leakage and ROS. The experimental design (treatments, replicates and samples) is adequately described.

The information about the statistical analysis is not exhaustive (what about ANOVA ?), it reports only a post-hoc test and the software used but should be improved.

Line 308: “ …  into 3.78 L pots and …” pots filled with peat, coir or what else ?

Lines 310-311: “… The greenhouse was located at … photoperiod of 14h daylight”; what about the greenhouse structure (steel ? zinc-coated ? or …) and covering material (glass, PE, PMMA ?).

Lines 352-353: “… The percentage of germination was calculated … 7 days of imbibition.”. Did you transform the percentage data in angular values (arc sin transformation) before subjecting them to ANOVA ? Please specify.

Lines 404-405: please include more information (see comments above).

All the Figures clearly report the obtained data previously described in the Results section.

Figure 1: modify the caption as suggested “… the seed size. Box plots are means ± standard error. ** indicates significant …”.

Figure 2. modify the caption as suggested “… each treatment. Vertical bars are means ± standard error. Percentage data were transformed in angular values (arc sin transformation) before subjecting them to ANOVA. ** indicates significant …”.

Figure 3. modify as suggested “… each treatment. Vertical bars are means ± standard error. Percentage data were transformed in angular values (arc sin transformation) before subjecting them to ANOVA. ** indicates significant …”.

Figure 4. modify as suggested “… each treatment. Vertical bars are means ± standard error. * and ** indicates significant …”.

Figure 6. modify as suggested “… each treatment. Vertical bars are means ± standard error. ** indicates significant …”.

The literature cited includes most recent studies conducted on the related topic but it should be slightly improved (see my comments above for the Introduction and the Discussion).

Please add this article in the list of references (and update their order):

  1. Giordano M., Pannico A., Cirillo C., Fascella G., El-Nakhel C., De Pascale S., Rouphael Y. 2020. Influence of priming methods on seed germinability and transplants performance in six vegetable species. Acta Hort. 1296: p. 297-304.
  2. Huang, M., et al., Morphological and physiological traits of seeds and seedlings in two rice cultivars with contrasting early vigor. 2016: p. 1-7.
  3. Lu, X.-L., et al., Genetic dissection of seedling and early vigor in a recombinant inbred line population of rice. Plant Science, 2007. 172(2): p. 212-220.

Author Response

Reviewer 2

Dear Reviewer 2,

Thank you for your kind words, please find our responses to your specific concern below.  

  1. Lines 244-245: modify as suggested “Great reservoir can provide adequate nutrients especially in the early developmental stage without any photosynthesis, as observed in some vegetable species subjected to seed priming treatments (31)”.

Response: We have modified as suggested in the main manuscript (Line 254-255).

  1. The information about the statistical analysis is not exhaustive (what about ANOVA ?), it reports only a post-hoc test and the software used but should be improved.

Response: Student’s t test was used for all of the data analysis because we were interested in the statistically significant between the two maternal environments.

  1. Line 308: “ …  into 3.78 L pots and …” pots filled with peat, coir or what else ?

Response: Information was added to the main manuscript (Line 318)

  1. Lines 310-311: “… The greenhouse was located at … photoperiod of 14h daylight”; what about the greenhouse structure (steel ? zinc-coated ? or …) and covering material (glass, PE, PMMA ?).

Response: Information was added to the main manuscript (Line 319-320)

  1. Lines 352-353: “… The percentage of germination was calculated … 7 days of imbibition.”. Did you transform the percentage data in angular values (arc sin transformation) before subjecting them to ANOVA ? Please specify.

Response: we did not perform the ANOVA for statistical analysis in this study, because we believe that Student’s t test is good enough for examining pairwise significant difference between the two maternal environment treatments.

  1. Lines 404-405: please include more information (see comments above).

 All the Figures clearly report the obtained data previously described in the Results section.

Figure 1: modify the caption as suggested “… the seed size. Box plots are means ± standard error. ** indicates significant …”.

Figure 2. modify the caption as suggested “… each treatment. Vertical bars are means ± standard error. Percentage data were transformed in angular values (arc sin transformation) before subjecting them to ANOVA. ** indicates significant …”.

Figure 3. modify as suggested “… each treatment. Vertical bars are means ± standard error. Percentage data were transformed in angular values (arc sin transformation) before subjecting them to ANOVA. ** indicates significant …”.

Figure 4. modify as suggested “… each treatment. Vertical bars are means ± standard error. * and ** indicates significant …”.

Figure 6. modify as suggested “… each treatment. Vertical bars are means ± standard error. ** indicates significant …”.

Response: We have modified the figure legend as suggested by this reviewer, except the description on ANOVA that was not performed in this study. For the caption description of each figure, we believe that our original description is more informative. For example, “Figure 3. Germination response of petunia seeds from greenhouse or tent plants to dehydration, salinity, and high temperature treatments”, can directly reveal what treatments were tested for seed germination, when compared to the caption title “Figure 3. Germination response of petunia seeds from greenhouse or tent plants to each treatment”.

  1. The literature cited includes most recent studies conducted on the related topic but it should be slightly improved (see my comments above for the Introduction and the Discussion).

Please add this article in the list of references (and update their order):

  1. Giordano M., Pannico A., Cirillo C., Fascella G., El-Nakhel C., De Pascale S., Rouphael Y. 2020. Influence of priming methods on seed germinability and transplants performance in six vegetable species. Acta Hort. 1296: p. 297-304.
  2. Huang, M., et al., Morphological and physiological traits of seeds and seedlings in two rice cultivars with contrasting early vigor. 2016: p. 1-7.
  3. Lu, X.-L., et al., Genetic dissection of seedling and early vigor in a recombinant inbred line population of rice. Plant Science, 2007. 172(2): p. 212-220.

Response: Highlighted paper was cited and the order was updated.

Reviewer 3 Report

The submitted paper bring interesting knowledge about seed germination of Petunia x hybrida originated from plants growing in greenhouse and air conditioned chambers differ in temperature and light condition. Obtained seeds were subjected to external stressors (polyethylene glycol (PEG), sodium chloride (NaCl), high temperature, and abscisic acid (ABA)) to determine seed germination percentage and seedling vigour. Authors pointed out that based on experimental results the difference in germination and seedling vigour of seeds from two different maternal environments might be due to the epigenetic memory inherited from the mother plants.

 I would recommend the paper for publication in Plants after the following revisions:

Comments:

  1. In fig. 2 row 103, Fig. 3 row 127 author used the “…seven-days imbibition…” Imbibition is intake of water to seeds to start germination and can be divided into three phases (citation – Akhila, S., Puthur, T.J. Priming-Mediated Stress and Cross-Stress Tolerance in Crop Plants, 2020, Pages 303-316). I recommend use “germination”.

  1. For more precisely characterisation of germination the author can used further characteristic as germination potential, seeds vigour index, seedlings vigour index, seedlings length index according to Abdul-Baki A, Anderson J (1973) Vigor determination in soybean seed by multiple criteria. Crop Sci 630-633

  1. In all text, there is lots of errors in writing numeric values and mm, µM, h a.o. which can be corrected in rows 135, 136, 137, 138, 158, 169, 305, 307, 308, 310, 311, 314, 319, 345, 373.

  1. In the Results, in Fig. 2 and 4 in the picture A on x axis must be changed uM to µM and separate form numbers (also in Fig. 3 A 200 mM, 250 mM, Fig. 1 in both 2 mm, Fig. 4 E 100 mM, Fig. 5, Fig. 6).

  1. In all text in misspelled a.g. 20°C – correct 20 °C.

  1. In all text – the sureoxide anion in form O2- is incorrect, correct ˙O2¯.

  1. In the Discussion, is necessary to write an apostrof in rows 236, 240, 242, 248, 280, 379, 381? If yes, it must be uniform.

  1. In Material and methods: what were the conditions of germination? It is not mentioned in this chapter.

  1. Why were seed sizes measured only after overnight imbibition? Is this method common?

  1. In the row 391, years are not given by the authors name and these citations are not given in the Reference chapter.

  1. In conclusion, the whole manuscript must be read carefully and corrected as recommended.

Author Response

Reviewer 3

Dear Reviewer 3,

Thank you for the detailed suggestions, please find our responses to your specific concern below.  

  1. In fig. 2 row 103, Fig. 3 row 127 author used the “…seven-days imbibition…” Imbibition is intake of water to seeds to start germination and can be divided into three phases (citation – Akhila, S., Puthur, T.J. Priming-Mediated Stress and Cross-Stress Tolerance in Crop Plants, 2020, Pages 303-316). I recommend use “germination”.

Response: The germination percentage was recorded after 7 days of incubating seeds on agar plate. This incubation of seeds on the moisten papers is termed as imbibition, which has been extensively used by seed biologists in various high-profile research papers, therefore we will follow the common guideline for describing this process. 

  1. For more precisely characterisation of germination the author can used further characteristic as germination potential, seeds vigour index, seedlings vigour index, seedlings length index according to Abdul-Baki A, Anderson J (1973) Vigor determination in soybean seed by multiple criteria. Crop Sci 630-633

Response: Germination description was added (line 350-351). We thank this reviewer’s suggestion, the seed germination may be manifested by different formats including germination potential, seed vigor index etc, yet all of these formats are based on the germination percentage under the testing conditions. Germination percentage is one format that has been commonly used by seed biologist as well as other plant biologist with research focus on, say, developmental biology. In order to attract broad audience, we adapted the characterization format used by both seed biologists, ecologists and other plant biologists. In addition, the goal of this study is mainly focusing on the germination of petunia seeds harvested from different growing conditions under different germinating stresses.

  1. In all text, there is lots of errors in writing numeric values and mm, µM, h a.o. which can be corrected in rows 135, 136, 137, 138, 158, 169, 305, 307, 308, 310, 311, 314, 319, 345, 373.
  2. In the Results, in Fig. 2 and 4 in the picture A on x axis must be changed uM to µM and separate form numbers (also in Fig. 3 A 200 mM, 250 mM, Fig. 1 in both 2 mm, Fig. 4 E 100 mM, Fig. 5, Fig. 6).
  3. In all text in misspelled a.g. 20°C – correct 20 °C.
  4. In all text – the sureoxide anion in form O2- is incorrect, correct ˙O2¯.
  5. In the Discussion, is necessary to write an apostrof in rows 236, 240, 242, 248, 280, 379, 381? If yes, it must be uniform

Response: Errors were fixed in the main manuscript.

  1. In Material and methods: what were the conditions of germination? It is not mentioned in this chapter.

Response: The description can be found in the manuscript lines 347-351. “Seeds from both maternal environments were tested on 1% agar plate as the control or on 1% agar plate infused with stressors to mimic different abiotic stresses. Germination test was carried out under 16 h fluorescent white light and 8 h darkness photoperiod at room temperature (25 °C) except for the high-temperature treatment. Seed with radicle protrusion of at least 1mm was considered germinated.”

  1. Why were seed sizes measured only after overnight imbibition? Is this method common?

Response: the seed size was measured after overnight imbibition to minimize the difference caused by different water content between both types of seeds. 

  1. In the row 391, years are not given by the authors name and these citations are not given in the Reference chapter.

 Response: we have fixed this error.

  1. In conclusion, the whole manuscript must be read carefully and corrected as recommended.

Response: Thanks for your recommendation, we have thoroughly read and revised this manuscript as needed. 
